# Disparity Between Functional and Structural Recovery of Placental Mitochondria After Exposure to Hypoxia

**DOI:** 10.3390/ijms26072956

**Published:** 2025-03-25

**Authors:** Jonathan R. Sierla, Laia Pagerols Raluy, Magdalena Trochimiuk, Julian Trah, Mariam Petrosyan, Lis N. Velasquez, Udo Schumacher, Dominique Singer, Julia Heiter

**Affiliations:** 1Division of Neonatology and Pediatric Critical Care Medicine, University Medical Center Hamburg-Eppendorf, 20246 Hamburg, Germany; jonathan.sierla@stud.uke.uni-hamburg.de (J.R.S.);; 2Department of Pediatric Surgery, University Medical Center Hamburg-Eppendorf, 20246 Hamburg, Germany; 3Institute of Anatomy and Experimental Morphology, University Cancer Center Hamburg, University Medical Center Hamburg-Eppendorf, 20246 Hamburg, Germany; 4Division of Molecular Immunology und Gastroenterology, I. Department of Medicine, University Medical Center Hamburg-Eppendorf, 20246 Hamburg, Germany; 5Center for Translational Immunology (HCTI), University Medical Center Hamburg-Eppendorf, 20246 Hamburg, Germany

**Keywords:** IUGR, FGR, JEG-3 cells, hypoxia, mitochondrial function, mitochondrial content, flow cytometry

## Abstract

Intrauterine growth restriction (IUGR) affects 5–10% of pregnancies with placental hypoxia, playing a key role as a common pathophysiological pathway of different etiologies. Despite the high metabolic rate of the placenta and its “gatekeeper” role in protecting the fetus from hypoxia, the response of placental mitochondria to hypoxic stress is not well understood. This study tested the hypothesis that transient exposure to hypoxia leads to a loss of placental mitochondria and affects their function. Human villous trophoblastic (JEG-3) cells were cultured under normoxic and hypoxic conditions for 24 h. Mitochondrial content was determined by flow cytometry before and after hypoxic exposure and after 24 h of normoxic recovery. Parameters of oxidative phosphorylation were assessed using a respirometric analyzer before hypoxic exposure and after normoxic recovery. Mitochondrial content decreased significantly from 88.5% to 26.7% during hypoxic incubation. Although it had increased to 84.2% after 24 h of normoxic recovery, oxidative phosphorylation parameters were still significantly suppressed to 1/2 to 1/3 of the pre-incubation levels. The results underscore the ability of placental cells to adapt mitochondrial content to O_2_ supply. Despite rapid recovery under normoxia, respiratory function remains suppressed, which may result in persistent impairment of adenosine triphosphate (ATP)-dependent synthetic and transport functions.

## 1. Introduction

Intrauterine growth restriction (IUGR), also called fetal growth restriction (FGR) [1], affects 5–10% of pregnancies. The causes are manifold, reflecting an interaction of maternal, fetal, placental, and genetic factors. The end result of the various pathologies is thought to be an imbalance between fetal nutritional requirements and placental nutrient supply, resulting in restricted fetal growth and permanent metabolic changes [2,3]. Furthermore, IUGR is a common cause of perinatal morbidity and mortality, leading to an increase in postnatal complications such as neonatal asphyxia, myocardial dysfunction, and necrotizing enterocolitis [4]. Long-term sequelae include impaired neurological development, type 2 diabetes, metabolic syndrome, cardiovascular diseases, and osteoporosis [3]. As the causes of IUGR vary widely, it remains unclear whether they share a common pathophysiological pathway. Basically, IUGR can be divided into cases with underlying pathological changes in placental structure [5], such as pre-eclampsia [6], and cases without primary structural changes, such as high-altitude pregnancies associated with hypobaric hypoxia [7]. However, both pathophysiological conditions eventually result in a reduced supply of oxygen to the placenta [7]. Under normal circumstances, it is estimated that only about 60% of the oxygen supply to the placenta is passed on to the fetus, while the placenta itself consumes nearly half of the overall oxygen supply [8]. In the event of a reduced supply, the placenta seems to be able to reduce its own oxygen consumption in favor of maintaining a sufficient supply to the fetus [7]. The placenta thus acts as a “gatekeeper” in the allocation of oxygen, and therefore energy [8]. Although the reduction in oxygen consumption partially compensates for the lack of oxygen, it is accompanied by a decrease in mitochondrial adenosine triphosphate (ATP) production, which in turn leads to impaired nutrient transport to the fetus [7]. It has previously been shown that placental tissue from IUGR pregnancies has a generally higher mitochondrial content, whereas the mitochondrial content in specific cell types such as cytotrophoblasts is decreased [9]. Mitochondrial function in cells with an altered mitochondrial content has not been extensively studied in placental tissue. We therefore used JEG-3 cells as a placental model [10,11,12] to investigate mitochondrial content and function after exposure to hypoxia. We hypothesized that transient exposure of placental JEG-3 cells to hypoxia leads to loss of placental mitochondria and affects their function.

## 2. Results

### 2.1. Mitochondrial Content in Normoxically and Hypoxically Cultured JEG-3 Cells

The mitochondrial content of JEG-3 cells in normoxic culture is about 88.5% ± 9.6% SEM, meaning that 88.5% of the cells contain viable mitochondria, as measured by flow cytometry. Dead cells after hypoxic incubation were not considered for calculation. After 24 h of hypoxic incubation, the content decreased to an average of 26.7% ± 9.9% SEM, (*n* = 6). After a recovery period of 24 h under normoxic conditions, it increased again to 84.2% ± 13.9% SEM on average. The differences were significant (*p* < 0.05) when comparing the mitochondrial content before and after hypoxia as well as the mitochondrial content after hypoxia compared to 24 h later under normoxic conditions (Figure 1).

### 2.2. Fluorescence Microscopy of Normoxically and Hypoxically Cultured JEG-3 Cells

The differences in the mitochondrial content can be visualized by fluorescence microscopy (Figure 2).

The samples measured immediately before hypoxic incubation had a significantly higher fluorescence intensity, analyzed by mean grey values, compared to samples measured after 24 h of hypoxic incubation (*n* = 6; *p* < 0.05). After 24 h of recovery under normoxic conditions, the samples regained their fluorescence, although the difference was not statistically significant compared to the fluorescence after hypoxic incubation (*p* = 0.07) (Figure 3).

### 2.3. OCR in Normoxically and Hypoxically Cultured JEG-3 Cells

In contrast to the fairly recovered mitochondrial content, the OCR of hypoxically cultured JEG-3 cells was still significantly lower compared to normoxically cultured JEG-3 cells when measured using the Seahorse XFe96 analyzer (Figure 4). Normoxically cultured JEG-3 cells consistently exhibited two-to-three-fold higher baseline (*p* < 0.01), ATP synthase activity (*p* < 0.01), maximal respiration (*p* < 0.01), extramitochondrial respiration (*p* < 0.01), spare respiratory capacity (*p* < 0.05), and proton leak (*p* < 0.01), compared to hypoxically cultured JEG-3 cells, although the latter had already undergone the 24 h recovery period under normoxic conditions (Figure 5).

## 3. Discussion

### 3.1. Decrease in Mitochondrial Content After Exposure to Hypoxia

IUGR appears to be associated with reduced oxygen supply to the placenta [7]. We therefore studied the mitochondrial content and OCR of JEG-3 cells under hypoxic compared to normoxic conditions. A transient decrease in mitochondrial content was observed after exposing the cells to hypoxia for 24 h. In line with our results, Vangrieken et al. found a significantly lower mitochondrial content in BeWo choriocarcinoma cells after exposure to hypoxia. OCR was not examined, yet a higher glycolysis rate was reported. They also demonstrated similar results in villous explants from human placentas exposed to hypoxia [13]. The decrease in mitochondrial content can be interpreted as a comparatively rapid down-regulation of mitochondrial respiration in response to reduced oxygen availability, possibly reflecting the placenta’s ability to reduce its own oxygen consumption in favor of maintaining oxygen delivery to the fetus. The active, regulatory nature of the reduction in mitochondrial content is supported by the finding of increased mitophagy in native placentas suffering from FGR [14].

### 3.2. Restoration of Mitochondrial Content After Re-Exposure to Normoxia

Another indicator of the adaptive nature of the variation in mitochondrial content is the nearly complete recovery observed after re-exposure to normoxic conditions. However, it is questionable whether this is a specific feature of human villous trophoblasts or whether it is due to the fact that JEG-3 cells are derived from choriocarcinoma. In the latter case, the rapid restoration of mitochondrial content after reoxygenation may reflect the increased metabolic activity associated with the ability of malignant cells to proliferate in an uncontrolled manner and to overcome hypoxic and nutrient-poor conditions [15,16]. On the other hand, a similarity between tumorous and placental tissue is sometimes assumed, owing to their common ability for invasive growth [17]. This may also be reflected in similar metabolic properties.

### 3.3. Persisting Functional Impairment of JEG-3 Cells After Exposure to Hypoxia

Although mitochondrial content recovers quickly, respiratory activity of transiently hypoxic JEG-3 cells is reduced compared to normoxic cells. Even after 24 h of re-exposure to normoxia, they exhibit a reduced baseline OCR, a reduced maximal respiration rate, and a reduced spare respiratory capacity compared to normoxic controls which, in this case, appears to be due to a functional suppression rather than a numerical reduction in mitochondrial activity. As a result, intermittent hypoxia may lead to at least a transient, perhaps persistent, impairment of energy turnover, independent of the mere presence of mitochondria in placental tissue. Notably, even proton leak as well as nonmitochondrial respiration and ATP production are significantly reduced in transiently hypoxic compared to normoxic JEG-3 cells (Figure 5). This suggests that the depletion and subsequent restoration of mitochondrial content, despite the persistent impairment of respiratory activity, is not accompanied by increased uncoupling rates. A lower proton leak at a lower OCR indicates a more efficient ATP production, possibly in favor of a rapid resumption of energy-dependent enzymatic and transport processes in placental tissue.

All in all, our study seems to confirm a fairly rapid decline in mitochondrial activity (both numerical and functional) of placental cells in response to hypoxia, possibly reflecting an adaptive reduction in autochthonous placental OCR. In addition, a dissociation between mitochondrial content and respiratory activity was found, with suppression of ATP production outlasting the relatively rapid numerical recovery of mitochondrial content. Further studies should investigate whether the ongoing suppression of respiratory activity results from a persistent passive impairment of mitochondrial function or reflects a sustained active switch in placental metabolism, e.g., towards anaerobic glycolysis. Furthermore, it would be worthwhile to investigate the extent to which respiratory activity can be restored, as well as how the mitochondrial number and function are affected by repeated or even permanent hypoxia, in order to better understand the limits of placental tissue adaptability to oxygen deprivation.

### 3.4. Limitations

Our study has some limitations. First, as mentioned above, the origin of JEG-3 cells from choriocarcinoma may affect their responses to hypoxia, due to several metabolic peculiarities such as upregulated glycolytic activity [16]. Nevertheless, JEG-3 cells have previously been used as a cellular model to investigate placental disorders and metabolism in various studies [10,11,12]. Second, measurement of OCR immediately after hypoxic incubation was not feasible due to the attachment time of 24 h after seeding. Furthermore, the determination of glycolytic (lactate production) rate in parallel with OCR was not part of the experimental protocol, although several studies have shown a higher glycolytic rate in placental (as well as malignant) cells exposed to hypoxia [7,13,18,19]. Third, the hypoxic incubation of JEG-3 cells at approximately 4% O_2_ results in an O_2_ partial pressure of about 30 mmHg, which corresponds to the mean O_2_ partial pressure in the fetal circulation [8]. Although the O_2_ tension on the maternal side of the placental barrier is higher, it must be considered that the experimental conditions may imply some degree of hyperoxia rather than normoxia.

## 4. Materials and Methods


### 4.1. Cell Culture and Experimental Protocol

JEG-3 cells are a human villous trophoblastic cell line derived from choriocarcinoma and obtained from the Leibniz Institute DSMZ, German Collection of Microorganisms and Cell Cultures (Braunschweig, Germany). They were cultured in Dulbecco’s modified Eagle’s Medium (DMEM; Gibco, Waltham, MA, USA) supplemented with 10% fetal bovine serum (FBS; Gibco, Waltham, MA, USA) and 1% penicillin–streptomycin (10,000 U/mL penicillin and 10 mg/mL streptomycin; PAN-Biotech, Aidenbach, Germany). Cells were maintained in a Panasonic MCO-19MUV-PE (Kadoma, Japan) cell incubator at 37 °C and 5% CO_2_ and used for experiments between passages 14 and 31. The medium was changed every 48–72 h. Cell splitting was performed at an approximate confluence of 80–90% using Accutase detachment solution (Capricorn Scientific, Ebsdorfergrund, Germany). Cell counting was performed using a Neubauer counting chamber (Merck Millipore, Darmstadt, Germany) with trypan blue stain (0.4%; Gibco, Waltham, MA, USA). For some experiments, JEG-3 cells were incubated in a hypoxia incubator chamber (Stemcell Technologies, Vancouver, BC, Canada) under hypoxic conditions at 37 °C and 4% O_2_ for 24 h. Oxygen was replaced by nitrogen and carbon dioxide in a 2:1 ratio within the hypoxia chamber. Mitochondrial content varies in JEG-3 cells; therefore we used samples with high and low mitochondrial content to perform pre-tests as a proof of concept for our experiments, showing that oxygen consumption rate (OCR) is reduced in JEG-3 cells with low mitochondrial content compared to those with high mitochondrial content. The main experiment aimed to analyze the difference in mitochondrial content and function for normoxically and hypoxically cultured JEG-3 cells. Therefore, half of the JEG-3 cell population was incubated under hypoxic conditions, while the other half was kept under normoxic conditions for 24 h prior to seeding on the Seahorse XFe96 cell culture microplate. The mitochondrial content of the hypoxically and normoxically cultured JEG-3 cells was then determined by flow cytometry at the same time as seeding. After seeding both samples, the cell culture microplate was kept in the incubator under normoxic conditions for approximately 24 h to obtain cellular attachment. The remaining cells from both samples were replaced in each flask and also kept in the incubator under normoxic conditions. The following day, Seahorse assay was run to compare the OCR of normoxically and hypoxically cultured JEG-3 cells. In addition, mitochondrial content of the two groups of JEG-3 cells was compared by flow cytometry 24 h after recovery under normoxic conditions (Figure 6).

### 4.2. Flow Cytometry and Fluorescence Microscopy

For the quantification of the mitochondrial content of the JEG-3 cells, flow cytometry was performed on a MACSQuant 10 (Miltenyi Biotec, Bergisch Gladbach, Germany). To detect mitochondria, JEG-3 cells were trypsinized, stained with 180 nM of MitoTracker^TM^ Green (In vitrogen, Waltham, MA, USA), and incubated for 30 min. After incubation, the cells were washed with phosphate-buffered saline (PBS; Gibco, Waltham, MA, USA) and resuspended in fluorescence-activated cell sorting buffer consisting of PBS with 1% bovine serum albumin (Gibco, Waltham, MA, USA) and 0.05% sodium azide (Thermo Fisher Scientific, Waltham, MA, USA). Prior to measurement, propium iodide (PI; Miltenyi Biotec, Bergisch Gladbach, Germany) was added to the samples to ensure cell viability. Finally, samples were measured on the MACSQuant and analyzed using FlowLogic^TM^ software, version 8.7 (Inivai Technologies, Mentone, Victoria, Australia). Flow cytometry was performed on cells cultured in normoxia and hypoxia to determine the amount of fluorescent mitochondria for each sample. Cells subjected to hypoxia were analyzed at three time points: before hypoxic incubation, immediately after 24 h of hypoxic incubation, and a further 24 h later while recovering under normoxic conditions. Normoxically cultured cells were analyzed immediately before seeding. Visual assessment of mitochondrial content was performed by digital imaging with Zeiss ZEN version 1.4.20 (Oberkochen, Germany) under light microscopy using a Zeiss Axio Vert.A1 (Carl Zeiss AG, Oberkochen, Germany). For this purpose, JEG-3 cells were stained with MitoTracker^TM^ Green as described previously. In addition, 4^′^,6-diamidino-2-phenylindole (DAPI; In vitrogen, Waltham, MA, USA) staining was performed to visualize the nuclei. The stained JEG-3 cells were then washed with PBS, resuspended in PBS, pipetted onto an object plate, and analyzed directly under the fluorescence microscope. ImageJ software, version 1.53K, (Bethesda, MD, USA) was used for fluorescence quantification. The mean grey value was measured and statistically analyzed.

### 4.3. Analysis of Oxidative Phosphorylation Activity

Cellular OCR was analyzed using the Seahorse XFe96 analyzer (Agilent Technologies, Inc., Santa Clara, CA, USA) to assess the difference in mitochondrial respiration of differently treated JEG-3 cells according to the mitochondrial (Mito) stress test protocol [20]. Materials for experiments using the Seahorse XFe96 analyzer were purchased from Agilent Technologies, Inc. (Santa Clara, CA, USA) with the exception of electron transport chain modulators. Oligomycin and antimycin A were supplied by Sigma-Aldrich (St. Louis, MO, USA). Carbonyl cyanide-p-trifluoromethoxyphenylhydrazone (FCCP) and rotenone were obtained from Cayman Chemical (Ann Arbor, MI, USA). The OCR of hypoxically cultured JEG-3 cells was determined in comparison to normoxic cells. The optimal cell density was determined to be 7.5 × 10^4^ cells/well. After seeding the cells on the Seahorse XFe96 cell culture microplate, the latter was placed in a normoxic incubator at 37 °C and 5% CO_2_ for a further 24 h, which is necessary to ensure adherence of the seeded cells to the bottom of the well. The Mito stress test was performed on the day after seeding according to protocol [20]. Only the oligomycin concentration was adjusted to 2 µmol per well for adequate measurement. Lower concentrations did not cause any deflection. The parameters to be assessed were baseline, reflecting unaffected mitochondrial respiration, without any manipulation of the electron transport chain; ATP-linked respiration after the addition of oligomycin (a complex V inhibitor); maximal respiration due to the injection of FCCP (an uncoupler of oxidative phosphorylation); nonmitochondrial respiration after the administration of a mixture of rotenone (a complex I inhibitor) and antimycin A (a complex III inhibitor); proton leak, which is a sign of mitochondrial damage or a mechanism used to regulate mitochondrial ATP production; and spare respiratory capacity, which is the ability of cells to respond to an energy demand [20]. Proton leak and spare respiratory capacity are not apparent from the Seahorse graphs. To obtain the OCR for proton leak, the OCR after rotenone and antimycin A injection is subtracted from the OCR after oligomycin injection. For spare respiratory capacity, the baseline OCR must be subtracted from the OCR after FCCP injection [20]. The assay was performed three times, four experiments were included per assay with three measurements per modulator, resulting in *n* = 36 measurements per modulator.

### 4.4. Statistical Analysis

GraphPad Prism v.8 (GraphPad Software, La Jolla, CA, USA) was used for statistical analysis. Data from Seahorse, flow cytometry, and fluorescence microscopy were analyzed by the unpaired *t*-test. Variations in mitochondrial content were analyzed by one-way ANOVA with Turkey’s test for post hoc analysis. All data are presented as mean ± standard error of mean (SEM) and significance was considered as *p* < 0.05.

## 5. Conclusions

In a combined flow cytometric and respirometric approach, human trophoblast (JEG-3) cells responded to transient hypoxia with a marked, rapidly reversible decrease in mitochondrial content that was outlasted by a persistent functional suppression of respiratory chain activity. However, it remains unclear whether this reflects a persistent impairment of mitochondrial function or a transient redirection of energy metabolism towards anaerobic sources.

## Figures and Tables

**Figure 1 ijms-26-02956-f001:**
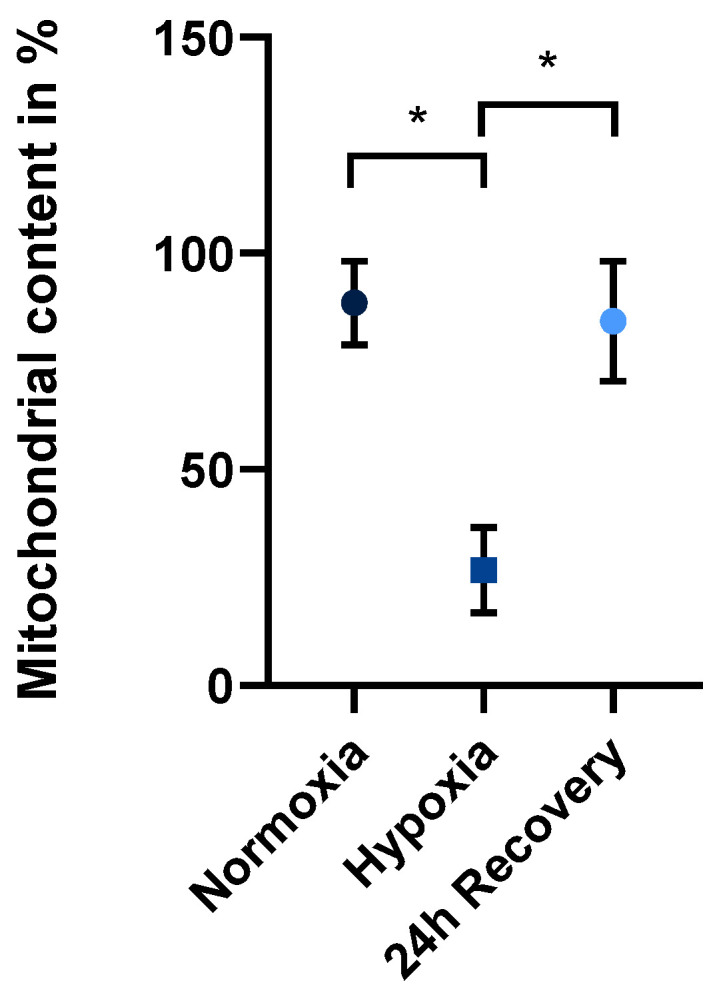
Mitochondrial content of JEG-3 cells in normoxia, at the end of 24 h hypoxia and after 24 h of normoxic recovery (*n* = 6; mean ± SEM; * *p* < 0.05).

**Figure 2 ijms-26-02956-f002:**
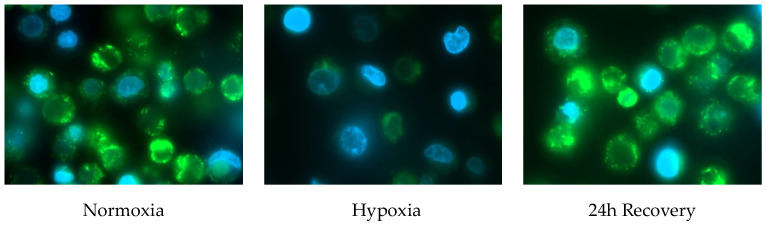
Fluorescence analysis of JEG-3 cells at the different time points of the experimental protocol. Blue staining represents nuclei; green staining represents mitochondria.

**Figure 3 ijms-26-02956-f003:**
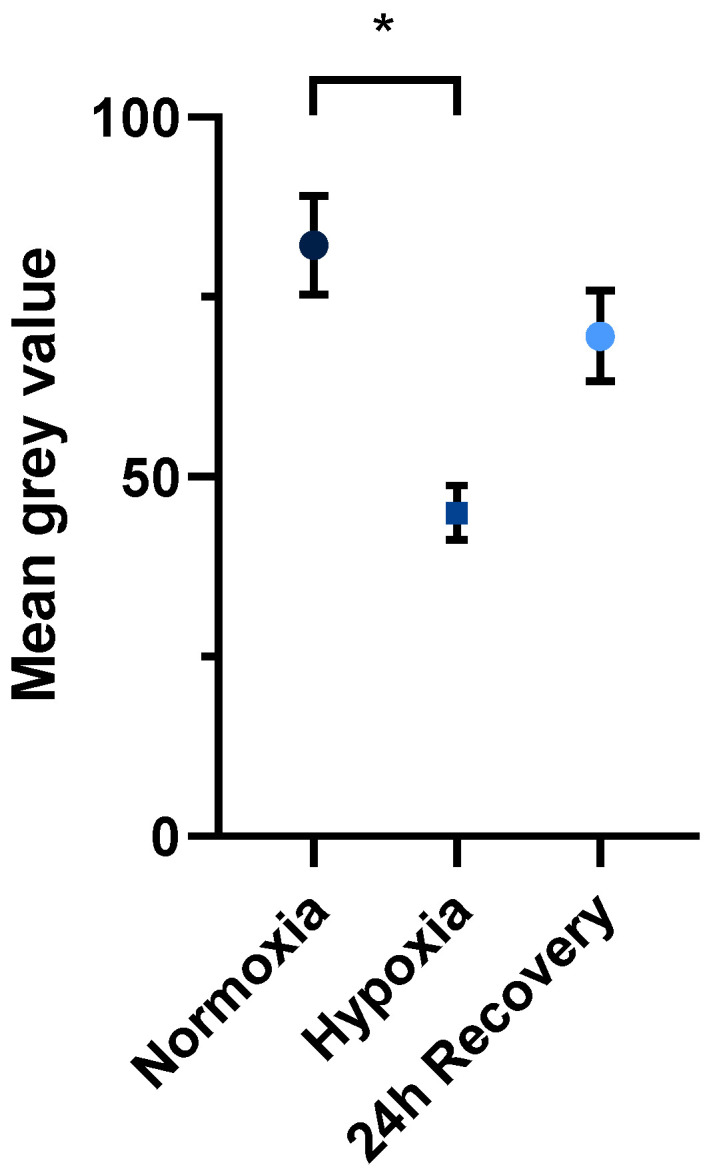
Statistical analysis of the mean grey value of JEG-3 cells at the different time points of the experimental protocol (*n* = 6; mean ± SEM; * *p* < 0.05).

**Figure 4 ijms-26-02956-f004:**
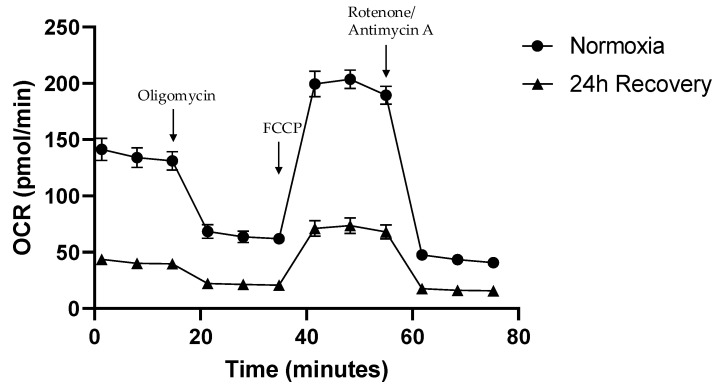
Oxygen consumption rate (OCR) over time according to the Seahorse Mito Stress Test protocol for posthypoxic (after 24 h of normoxic recovery) vs. normoxic JEG-3 cells (FCCP = carbonyl cyanide-p-trifluoromethoxyphenylhydrazone) (*n* = 4; mean ± SEM).

**Figure 5 ijms-26-02956-f005:**
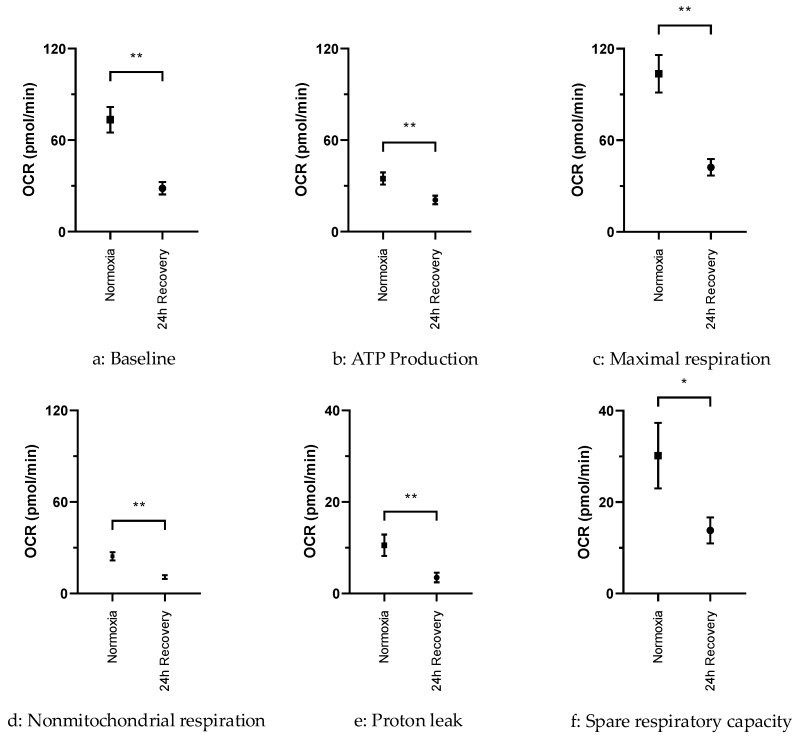
Statistical comparison of the different parameters of oxidative phosphorylation in posthypoxic (after 24 h of normoxic recovery) vs. normoxic JEG-3 cells (*n* = 36; mean ± SEM; * *p* < 0.05, ** *p* < 0.01). (**a**) Baseline—OCR without any mitochondrial manipulation of electronic transport chain. (**b**) ATP production—OCR after oligomycin injection. (**c**) Maximal respiration—OCR after FCCP injection. (**d**) Nonmitochondrial respiration—OCR after rotenone and antimycin A injection. (**e**) Proton leak—ATP production minus nonmitochondrial respiration. (**f**) Spare respiratory capacity—Maximal respiration minus baseline.

**Figure 6 ijms-26-02956-f006:**

Timescale of the experimental protocol.

## Data Availability

The data presented in this study are available on request from the corresponding authors.

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
