# Peer review of "Disparity Between Functional and Structural Recovery of Placental Mitochondria After Exposure to Hypoxia"

_ijms, 2025, doi:10.3390/ijms26072956_

Round 1

Reviewer 1 Report

Comments and Suggestions for Authors

It is well known that hypoxia leads to a reduction in mitochondrial content, decreased levels of key molecules involved in the electron transport chain and increased expression and activity of glycolytic enzymes. Here, the authors use a cell line (JEG-3) as a placental model to investigate mitochondrial content and function by exposure to hypoxia and to assess mitochondrial recovery. They found a disparity between the functional (no recovery) and structural recovery of placental mitochondria after exposure to hypoxia.
The topic is very interesting from a clinical point of view, as placental hypoxia is a common disorder that can affect nutrition and energy transfer to the fetus, leading to restricted fetal growth.

The manuscript is clear, concise and well written. The methodology is well described, although precise statistical information is lacking. Overall, the paper has two weaknesses, which are described below.

First of all, a trophoblast tumour line model does not seem to be the most appropriate to study energy metabolism. Have the authors considered using placental explants, which are a more physiological model?

Secondly, it is a very short article. Few experiments were carried out. We would have expected to see a study of cellular oxidative stress and ROS production, or HIF-1α levels and the molecular pathways that control mitochondrial content and functional recovery.

Minor comments

Methods. The cells used correspond to the passages between 14 and 31, Aren't they too old?
Lines 67-68: "Oxygen was replaced by nitrogen and carbon dioxide in a 2:1 ratio in the hypoxia chamber"; what does it mean? are the cells in hypercapnia?
Results. This section is not very clear. It is necessary to clarify the legend of the figures and to include the number of experiments (n). For example, what does % mitochondrial content mean? How is it calculated? Please clarify.
References. The text for all references is repeated. Please, correct it.

Author Response

Thank you very much for taking the time to review this manuscript. Please find the detailed responses below and the corresponding revisions/corrections highlighted in the re-submitted files.

Comment 1: First of all, a trophoblast tumor line model does not seem to be the most appropriate to study energy metabolism. Have the authors considered using placental explants, which are a more physiological model?

Response 1: Thank you for pointing this out. We agree with this comment. However, we decided to start with in vitro research using a cell line that has been served as a placenta model in different studies before. We are aware of the fact that a tumor cell line might not be the most appropriate model to study energy metabolism, although placenta cells and tumor cells have a lot in common. Next, we will use human villous trophoblasts before proceeding to native placental cells (the selection and isolation of which, of course, is not easy).

Comment 2: Secondly, it is a very short article. Few experiments were carried out. We would have expected to see a study of cellular oxidative stress and ROS production, or HIF-1α levels and the molecular pathways that control mitochondrial content and functional recovery.

Response 2: We totally agree. Nevertheless, we focused on the Seahorse assay to analyze mitochondrial function after hypoxic stress as this has not been done before. For further research, ROS production and HIF-1α levels should be investigated in our model.

Minor Comments

Methods: The cells used correspond to the passages between 14 and 31, Aren't they too old?

Response: Given that most articles don’t classify passages, we assume that our cells were not too old, especially as their function seemed similar in repeated analyses. However, we did also find articles that used cells in similar passages, cf. reference 10.

Lines 67-68: "Oxygen was replaced by nitrogen and carbon dioxide in a 2:1 ratio in the hypoxia chamber"; what does it mean? are the cells in hypercapnia?

Response: Yes, the cells were in hypercapnia as the mixture used was based on the standard operating procedures in our laboratory to induce hypoxic stress.

Results: This section is not very clear. It is necessary to clarify the legend of the figures and to include the number of experiments (n). For example, what does % mitochondrial content mean? How is it calculated? Please clarify.

Response: We are sorry to hear that our results seem unclear. We have clarified the legend of figure 2, 4, and 6 by including the number of experiments. We have also added an explanation on how the mitochondrial content is calculated (page 4, line 148).

References: The text for all references is repeated. Please, correct it.

Response: Thank you for the remark, we have corrected it.

Reviewer 2 Report

Comments and Suggestions for Authors

This is a very interesting investigation, showing us changes in mitochondrial dynamics during hypoxia in placental tissue.

I have a few questions and suggestions.

Methods

Could you please explain what exactly you measured under the „baseline“ conditions for OCR?

Did you use rotenone and antimycin A as a mixture or separately?

Results

Were there any dead cells after the experiment?

Figure 3 Legend. Although there is an explanation in the Method section, could you provide some additional information about bluish/greenish staining in cells?

Figure 6 Legend. Please clarify (add) what pharmacological modulation and calculation were used for each figure. How did you measure ATP production?

Discussion

Line 200. You say that „intermittent hypoxia may lead to persistent impairment of energy turnover“. However, you have assessed the condition after 24 hours of recovery, so it is difficult to conclude whether cells will fully recover their OCR or if this is persistent. Maybe it would be better to limit the conclusion to the period tested.

Line 202. You note that „even proton leak.. are significantly reduced in transiently hypoxic compared to normoxic JEG-3 cells“. How do you explain a higher proton leak, as a measure of mitochondrial damage, in normoxic cells vs. hypoxically cultured cells? (Results 3.3. OCR in normoxically and hypoxically cultured JEG-3 cells)

Conclusion

I have to disagree with a part of your conclusion stating that transport and synthetic processes are due to the lower ATP production. Firstly, this is something that you did not assess in your study. It is better to reconsider the conclusion emphasizing your results. Secondly, ATP is not necessarily a limiting factor for cell proliferation.

Mitochondrial energetic metabolism is an intersection of cellular cycle regulation deciding between proliferation or quiescence. A sort of switch that affects cellular phenotype.

In cancers, aerobic glycolysis with increased glucose consumption facilitates the production of electron carriers and provides important biosynthetic precursors for anabolic reactions. As you mentioned, other studies found higher glycolytic rates in placental cells exposed to hypoxia.

Perhaps the delay in maximal OCR and ATP production is due to this divergence in metabolic regulation with signaling redirected toward synthetic processes and cellular recovery.

Author Response

Thank you very much for taking the time to review this manuscript. Please find the detailed responses below and the corresponding revisions/corrections highlighted in the re-submitted files.

Comment 1:

Methods

Could you please explain what exactly you measured under the „baseline“ conditions for OCR?

Response 1:

The “baseline” conditions measure oxygen consumption rate over time in JEG-3 cells without any mitochondrial manipulation of electron transport chain (page 4, line 127).

Comment 2:

Did you use rotenone and antimycin A as a mixture or separately?

Response 2:

We used a mixture of rotenone and antimycin A which was recommended in the Seahorse user manual (page 4, line 130).

Comment 3:

Results

Were there any dead cells after the experiment?

Response 3:

There were several dead cells after hypoxic incubation, nevertheless only viable cells were considered for calculations (page 4, line 149).

Comment 4:

Figure 3 Legend. Although there is an explanation in the Method section, could you provide some additional information about bluish/greenish staining in cells?

Response 4:

We have clarified the legend of figure 3 accordingly: Blue staining displays nuclei, green staining displays mitochondria.

Comment 5:

Figure 6 Legend. Please clarify (add) what pharmacological modulation and calculation were used for each figure. How did you measure ATP production?

Response 5:

We have clarified the legend of figure 6 including pharmacological modulations and calculations. ATP production is measured via Seahorse by mitochondrial stress test after injection of oligomycin (page 7, Figure 6).

Comment 6:

Discussion

Line 200. You say that „intermittent hypoxia may lead to persistent impairment of energy turnover“. However, you have assessed the condition after 24 hours of recovery, so it is difficult to conclude whether cells will fully recover their OCR or if this is persistent. Maybe it would be better to limit the conclusion to the period tested.

Response 6:

Thank you for your remark, we totally agree and changed our wording. Please refer to the new uploaded version of the article.

Comment 7:

Line 202. You note that „even proton leak. are significantly reduced in transiently hypoxic compared to normoxic JEG-3 cells“. How do you explain a higher proton leak, as a measure of mitochondrial damage, in normoxic cells vs. hypoxically cultured cells? (Results 3.3. OCR in normoxically and hypoxically cultured JEG-3 cells)

Response 7:

“Proton leak is a sign of mitochondrial damage or a mechanism to regulate the mitochondrial ATP production.”(page 4, line 131)

Answering your question, we addressed this topic in the discussion:

“Notably, even proton leak as well as nonmitochondrial respiration and ATP production are significantly reduced in transiently hypoxic compared to normoxic JEG-3 cells (Figure 6). This suggests that the depletion and subsequent restoration of mitochondrial content, despite the persistent impairment of respiratory activity, is not accompanied by increased uncoupling rates. A lower proton leak at a lower OCR indicates a more efficient ATP production, possibly in favor of a rapid resumption of energy-dependent enzymatic and transport processes in placental tissue.” (page 8, line 207)

Comment 8:

Conclusion

I have to disagree with a part of your conclusion stating that transport and synthetic processes are due to the lower ATP production. Firstly, this is something that you did not assess in your study. It is better to reconsider the conclusion emphasizing your results. Secondly, ATP is not necessarily a limiting factor for cell proliferation.

Mitochondrial energetic metabolism is an intersection of cellular cycle regulation deciding between proliferation or quiescence. A sort of switch that affects cellular phenotype.

In cancers, aerobic glycolysis with increased glucose consumption facilitates the production of electron carriers and provides important biosynthetic precursors for anabolic reactions. As you mentioned, other studies found higher glycolytic rates in placental cells exposed to hypoxia.

Perhaps the delay in maximal OCR and ATP production is due to this divergence in metabolic regulation with signaling redirected toward synthetic processes and cellular recovery.

Response 8:

Thanks for the valid point. We have shortened and rephrased our (discussion and) conclusion accordingly.

Round 2

Reviewer 1 Report

Comments and Suggestions for Authors

The authors have adequately addressed my concerns. The presentation of the manuscript has improved.